# Mapping Iodine Sources for Human Nutrition in Portugal Considering Geography, Seasonality, and Processing: Milk and Plant-Based Milk Alternatives

**DOI:** 10.3390/nu17223606

**Published:** 2025-11-18

**Authors:** Sarai Isabel Machado, Ana Machado, Adriano A. Bordalo, Susana Roque, Nuno Borges, Joana Almeida Palha

**Affiliations:** 1Life and Health Sciences Research Institute (ICVS), School of Medicine, University of Minho, 4710-057 Braga, Portugal; 2Laboratory of Hydrobiology and Ecology, Institute of Biomedical Sciences (ICBAS-UP), University of Porto, 4050-313 Porto, Portugal; ammachado@icbas.up.pt (A.M.);; 3Interdisciplinary Centre of Marine and Environmental Research (CIIMAR), University of Porto, 4450-208 Matosinhos, Portugal; 4Faculty of Nutrition and Food Sciences, University of Porto, 4150-180 Porto, Portugal

**Keywords:** dairy, plant-based drinks, iodine, fortification

## Abstract

Background: Iodine is a micronutrient essential for the synthesis of thyroid hormones and crucial throughout life. Milk is potentially one of the major contributors to iodine intake in many countries, including Portugal, due to its consumption patterns. Objectives: This study characterizes iodine content seasonality in milk and plant-based milk alternatives commercially available in Portugal. Results: Milk products representing summer and winter seasonal pastures and plant-based alternatives were obtained from major Portuguese supermarkets. Iodine was quantified by the Sandell–Kolthoff reaction in 146 winter and 142 summer milk products, as well as in 128 plant-based alternatives. Cow’s milk contained relevant iodine levels (Md = 19.9, IQR = 9.9 µg/100 mL), with no influence of thermal processing, fat or lactose content, season, or being organic and/or from pasture. However, regional differences were observed. The iodine concentration in plant-based drinks was residual, except for four fortified products. Conclusions: This study provides evidence that milk is still a relevant source of iodine in Portugal, whereas most plant-based alternatives provide negligible iodine, unless fortified. Given shifts towards plant-based diets, monitoring iodine sources and adjusting health policies to fulfil nutritional requirements are pivotal to ensure iodine adequacy.

## 1. Introduction

Iodine is crucial for the synthesis of thyroid hormones, which regulate numerous metabolic pathways throughout life and are essential for normal central nervous system development [1,2,3]. As such, it is a public health priority to prevent iodine deficiency disorders [1] and associated neurodevelopmental impairment. The World Health Organization identifies universal salt iodization as the most effective and cost-efficient measure for that purpose, since it is easy to implement and adjust to salt-intake reduction policies [4,5].

Diet is the primary iodine intake source, and the iodine content in foods can vary with environmental factors such as soil and water characteristics [6]. The foods with the highest iodine contents are fish, seafood, and some species of seaweed. Iodine is also present, at moderate concentrations, in milk and dairy products. The European Union regulation considers a food product being a source of iodine when the iodine content is equal to or higher than 15% of the nutrient reference value per 100 mL or per serving, which is the case for milk [7,8]. Given the frequent consumption of dairy in many diets, it often constitutes a major contribution to iodine intake in many countries, including Portugal [9,10,11]. In the Porto region of Portugal, dairy was found to contribute to 55% of total iodine intake in working adults, which is 21% of the adequate daily iodine intake [11], and to be a relevant source in pregnant women [9]. As for milk itself, data from the latest National Food, Nutrition and Physical Activity Survey (IAN-AF), 2015–2016, points to milk consumption of 149 g/day by adults, which would correspond, in accordance with the scientific update on the iodine content of Portuguese foods, to a 24% contribution to the recommended iodine intake [12,13]. Interestingly, when considering the Portuguese Food Balance Sheet, milk’s representativity in terms of total food availability decreased 10.6% from the period 2016–2020 to 2020–2024, which points to a change in dietary habits [14].

Mechanistically, food iodine reflects environmental iodine in soil and water, while milk iodine additionally depends on feed supplementation, on-farm hygiene practices (e.g., iodophor use), and dietary goitrogens along the dairy supply chain [15,16,17]. Worldwide, there is increasing concern about the impact of diets on sustainable living and a trend toward adopting plant-based alternatives to commonly used animal products. On the one hand, dietary recommendations increasingly incorporate sustainability and planetary health considerations, including a reduction in animal products and prioritization of plant-based foods, as highlighted by the EAT–Lancet Commission [18]. On the other hand, these shifts raise concerns regarding micronutrient deficiencies, as in the case of iodine in plant-based milk alternatives [19,20]. In Portugal, iodine fortification of plant-based drinks remains voluntary, and its coverage has not been reported. In addition, iodine content information on product labels is inconsistently provided, further motivating surveillance of dietary iodine sources. Altogether, it is important to map and monitor iodine-rich foods, fortification, and supplementation strategies to adjust public health policies and food fortification programs accordingly and prevent iodine deficiency disorders.

The present study comprehensively characterized the iodine content in milk, comparing seasonality and processing characteristics, and plant-based milk alternatives available for consumers from major retailers in Portugal. Furthermore, it intended to estimate the contribution of milk to iodine intake.

## 2. Materials and Methods

### 2.1. Milk and Plant-Based Alternatives Collection

One unit of every milk and plant-based milk alternative available from the major Portuguese retail chains (on site from Continente, Pingo Doce, Auchan, Intermarché, Lidl, Aldi, Mercadona, El Corte Inglés, and E.Leclerc, and online from 360 hyper) was seasonally collected.

In the case of milk, to characterize summer pasture, collection was performed in October and November of 2023 (*n* = 142), and to characterize winter pasture, collection was performed in March and April of 2024 (*n* = 146) (considering the typical production-to-distribution lag period of three months).

Cow, sheep, and goat milk samples were included, as well as milk specifically labelled for children 1–3 years of age. Flavoured milk products (e.g., chocolate or cereals) were excluded. For plant-based milk alternatives (*n* = 128), all unflavoured and flavoured variants were included (soy, oat, almond, rice, hazelnut, coconut drinks, and combinations).

All label information was recorded: processing (UHT/pasteurized), fat level, lactose-free status, organic/pasture-fed, nutritional panels, origin/batch codes, and any iodine-related information. Country and region were attributed through the plant code of identification. Organic milk and/or milk from pasture were considered when reporting pasture-fed or biologic (organic) attributions.

Milk products were stored according to label guidance, UHT products at ambient temperature and pasteurized products under refrigeration. Plant-based milk alternatives (all UHT and shelf-stable products) were kept at room temperature. All items remained in their original, unopened packaging and were analysed within their labelled shelf life (before the stated best-before date). Prior to opening, containers were inspected for integrity (no swelling, leakage, or damaged seals). Immediately beforehand, containers were gently inverted to homogenize the contents.

### 2.2. Iodine Analysis

The Sandell–Kolthoff kinetic colorimetric method has been widely used for iodine determination in milk and dairy products [21,22,23], and was selected for its accessibility and throughput in routine monitoring laboratories. Iodine content was assessed by a matrix-adapted procedure [24,25], following Machado et al. [24,25]. Samples were digested with 1 M ammonium persulfate digestion at 100 °C in a water bath for 1 h. After reaching room temperature, arsenious acid reagent (1 M H_2_SO_4_, 0.43 M NaCl, 0.05 M As_2_O_3_) was added at a 2.5:1 (*v*/*v*) reagent–digest–sample ratio and allowed to react for 15 min prior to the addition of cerium (IV) solution (76 mM; 1:11.5 (*v*/*v*) ratio relative to mixture solution). Colour development was read exactly 30 min after the addition of cerium (IV) at 420 nm. A standard addition approach was employed to correct for potential matrix effects. For each analytical run, a six-point external calibration curve in water (0–468 µg I/L; 0–46.8 µg I/100 mL) was prepared. In parallel, a matrix-matched standard addition curve using the same nominal iodine standards was constructed for every sample. The iodine concentration of each sample was obtained from the x-intercept of the sample standard addition regression. The method limit of detection (LOD) and limit of quantification (LOQ) were 0.96 µg I/100 mL and 1.66 µg I/100 mL, respectively. Within-run repeatability, estimated from duplicate pairs, was 2.74% relative standard deviation (RSD) for milk in the summer series and 3.03% RSD for milk in the winter series. For plant-based drinks the RSD was 3.96%. Between-batch precision, estimated from an in-house quality control material (pooled UHT cow’s milk aliquots) was analysed in every batch; the RSD across batches was 2.18%. The method accuracy was assessed using a commercial infant formula with a declared iodine content of 15 µg I/100 mL (treated as a practical reference sample). The recovery of this milk was 98 ± 2% (*n* = 8). In addition, 100 milk samples and 48 plant-based drink samples were spiked at low and mid-levels within the calibration range, yielding recoveries of 94–109%. Because milk and plant-based drinks contain lipids and proteins that can interfere with catalytic colourimetry, alternative pre-treatments reported for milk (e.g., acidic vanadate/perchloric acid; Hedayati et al., 2007) were considered [23]. Ammonium persulfate digestion was selected for fitness-for-purpose and reagent safety, and its adequacy is supported by the in-matrix validation data presented. However, very high-fat or strongly fortified specialty milks may require more aggressive pre-treatments; in such cases, verification of recovery and precision is recommended. The analyst was blinded to the product characteristics.

### 2.3. Statistical Analysis

Analyses were performed in IBM SPSS, 29.0 for Mac. Distribution assumptions were assessed using the Shapiro–Wilk test and the Kolmogorov–Smirnov test (considering the number of samples), skewness, kurtosis coefficients, and histogram analysis.

To perform comparisons, only cow’s milk was included to prevent the interference of species-related variability in iodine content with a small representation.

To explore relevant determinants in Portuguese milk, a quantile regression was conducted to evaluate the impact of thermic processing, lactose content, being organic, fat level, season, and region on iodine levels. Products of the same brand and with the same characteristics but from different seasons were treated as independent samples. A 95% confidence level was defined.

## 3. Results

### 3.1. Milk

Table 1 presents the iodine content of 142 summer and 146 winter milk samples. Among these, 114 cow’s milk products were available in the two seasonal collection periods.

Table 2, where the adjusted medians are presented, shows that processing does not affect the iodine milk content. Similarly, no difference was found with respect to the milk being organic and/or from pasture and considering the season (summer or winter). However, milk from the centre region showed a higher iodine content and milk from the southern mainland region and the Azores Islands showed lower iodine levels when compared to the northern and Lisbon regions.

Adjusted median values from the model are presented in Table 2.

Considering the winter collection (which represents the most recent market for all brands), 24% of the samples reported the iodine content on the nutrition label and 11% presented health claims regarding iodine (e.g., “contributes to normal growth of children”, “contributes to normal cognitive function”, and “contributes to normal energy-yielding metabolism”). In the 34 milk products reporting iodine contents, the median iodine content statement on food labels was 20.0 μg (IQR = 0.8), and the median measured iodine content was 23.6 μg (IQR = 14.4).

Winter iodine content according to region and country is presented in Table 3. Identical results were obtained for the summer iodine content.

Considering the milk intake of the Portuguese population, the guidelines for adequate iodine intake, and the average iodine content in the milk reported here, Table 4 shows that the estimated contribution of milk to iodine intake in Portugal is 58% for children and 20% for adults (Table 4) [12,26,27,28].

### 3.2. Plant-Based Milk Alternatives

Plant-based alternatives’ iodine contents, as well as their reported mineral fortification, are described in Table 5. Plant-based alternatives were grouped according to the type of drink. Coconut drinks present a median iodine content of 7.9 μg/100 mL (the highest) and hazelnut drinks of 1.8 μg/100 mL (the lowest). More than half of the products are fortified with calcium (60%) and vitamin D (56%), 32% with vitamin B12, 19% with vitamin B2, 12% with vitamin A, and 5% with iodine.

In the four plant-base drinks reporting iodine supplementation, the iodine content stated on the food labels was 22.5 μg, and the median measured iodine content was 20.3 μg (SD = 10.8).

Considering plant-based alternatives as a substitute for milk, following the same rationale as Table 4, with a median iodine content of 3.3 μg/100 mL, the estimated contribution to iodine intake would be 3% and 6% for adults and children, respectively.

## 4. Discussion

This study maps the iodine content of milk and plant-based milk alternatives available to the general consumer in Portugal. It confirms milk as a key dietary iodine source, regardless of thermal processing, fat level, or lactose-free status. On the contrary, unfortified plant-based alternatives contain residual amounts of this micronutrient. The absence of winter–summer seasonal differences in milk iodine content suggests that feeding and managing practices are similar throughout the year. Of note, even though soils are poorer in iodine than commercial animal feed (usually fortified with iodine), no difference was found between pasture and organic milk. The definition of “organic” implies that more than 60% of the feed must be fresh or conserved forage; still, the amount of animal feed used likely suppresses the differences [29]. The lower concentrations of iodine found in milk from the Azores and southern regions are in agreement with other studies [16,30]. The Azores Islands’ geomorphology, as well as climate characteristics, irregular orography, and rainfall, are known to decrease iodine environmental availability [31].

Overall, the iodine concentrations in milk available from Portuguese retailers are similar to those reported in other European countries (e.g., France and Spain) [32,33].

Several factors influence iodine levels in milk, in addition to the iodine concentration in feed [15,16]. Goitrogens, substances that inhibit iodine uptake by the thyroid and mammary glands, are present in several plants used in cattle feeding (e.g., canola, kale, soybean, millet, and linseed) and may lead to overestimation of iodine absorption. Another relevant farming practice is the use of disinfectants. Iodine-containing teat sanitizers are commonly employed and, through skin absorption, can contribute substantially to the iodine content in milk [34,35]. Nevertheless, the main contributor to iodine milk concentration remains the cattle feed in the diet. While there is no specific legislation regarding cattle feed in Portugal, inspection of feed labels for small and industrial farmers indicates that concentrates typically contain about 1.5 mg iodine/kg. Considering that this feed concentrate is complemented with forage such as silage and hay, the overall iodine content per dry matter is in accordance with National Research Council’s recommendations (0.5 mg/kg dry matter) [36] and remains well below the maximum of 5 mg/kg dry matter recommended by the European authorities [37]. On average, a cow ingests about 6–9 kg of commercial feed per day, and roughly 30% of ingested iodine is transferred into the milk (~9 mg iodine intake → 30% secretion efficiency → 14 μg/L). Therefore, this would account for about half of the iodine content present in the milk [38]. The remainder derives from forage and/or absorption of iodine-containing disinfectants commonly used in livestock practice [34]. Of note, cattle feed in Portugal is also supplemented with selenium, another essential element for thyroid hormone synthesis (deiodinase enzymes are selenium-dependent). Selenium deficiency can, therefore, exacerbate iodine deficiency [39].

Considering the current recommendations on adequate iodine intakes and milk consumption as assessed by the National Food, Nutrition and Physical Activity Survey (2015–2016), milk fulfils about 20% of iodine requirements for adults and older adults (similar to what was reported by Machado et al. [11]), and over 50% for children.

By contrast, the low iodine content of plant-based milk alternatives and the rarity of iodine fortification of these beverages raise concerns as plant-based choices expand because this shift may compromise adequate iodine intake. Similar results on the contribution of milk to iodine intake were found in Norway, the United Kingdom, and Ireland [40]. In addition, the proportion of iodine fortification in plant-based alternatives in these countries is also low [41,42].

The present study reinforces that most plant-based alternatives are not comparable to milk regarding iodine content and should not be considered nutritional substitutes in this dimension. It also calls for consideration on the potential need for fortification of plant-based drinks. A study in the United Kingdom using a dietary modelling approach demonstrated that replacing milk with a non-fortified plant-based product would decrease iodine intake by up to 58% in age groups where milk makes a higher contribution and would increase the proportion of individuals with iodine intake below the recommended levels across all age groups [43]. Our findings indicate that such a dietary shift in Portuguese eating habits would severely compromise the population’s iodine adequacy. The same study found that fortification of 22.5 µg/100 mL would be sufficient to prevent such a decrease [43].

Within the European Union, food fortification is voluntary but regulated, recognizing that micronutrients may be added to food for improvement of the population’s nutritional status and to ensure that food proposed as an alternative can offer comparable nutritional value [44]. While fortification with calcium and vitamin D is already common in many formulations, iodine should also be considered.

Although milk represents an important iodine-rich food, the milk industry seldom highlights this added value for healthy human nutrition. Only 24% of milk products presented the iodine content on the label, and even fewer used the iodine health claims accepted by the European authorities [45]. Inclusion of this information could increase public awareness of the nutritional value of iodine for health.

A final consideration concerns the iodine content of milk and the fact that children generally consume more milk than adults, which makes them more likely to achieve iodine sufficiency. The median urinary iodine concentration of school-aged children is often used as a proxy for population iodine status [19], but this assumption may need to be revisited. This is particularly relevant in countries such as Portugal, where no national iodine fortification program exists beyond the requirement for iodized salt in school canteens [3], a measure that is reportedly not fully implemented [4]. Moreover, iodized household salt accounts for only a small fraction of total salt sales (11% in 2023) and thus makes a minimal contribution to overall iodine intake [5]. These factors collectively underscore the critical importance of milk as a dietary source of iodine.

A strength of the present work is the broad coverage of the major national retailers assessed across seasons and the linkage of analytical results to product characteristics. The cross-sectional nature of the study, the retail-snapshot design with potential product turnover between collection, the lack of batch replication and limitations related to the measurement method itself, the limited representativeness of artisanal milk products (raw milk, fermented milk, or processed products were not included), the absence of direct linkage to individual consumption or urinary biomarkers, and the reliance on the limited on-label information of some products were limitations of the study. In addition, the contributions of milk to daily iodine intake will be revised as national milk intake data are currently being assesssed. From the recent national food balance sheet, milk intake may be lower and, as such, the contribution of milk to iodine intake smaller than that estimated here.

## 5. Conclusions

In summary, the present study maps the iodine content and variability of milk and plant-based beverage sources in Portugal. These findings provide scientific knowledge to support public health authorities to decide on the most adequate measures to ensure iodine sufficiency in the population, not only through the implementation of salt iodization but also through fortification of plant-based milk alternatives, since their iodine contents are low.

## Figures and Tables

**Table 1 nutrients-17-03606-t001:** Characterization of iodine content of milk products.

Type of Milk	Summer Iodine Content		Winter Iodine Content	
	*n*	Median (IQR) (μg/100 mL)	*n*	Median (IQR) (μg/100 mL)
**Cow’s milk**	132	20.2 (6.3)	137	19.9 (9.9)
Ultra-pasteurized	126	19.8 (6.4)	132	19.9 (9.5)
Pasteurized	6	25.0 (5.1)	5	29.0 (8.5)
**Sheep’s milk**	1	72.3	1	105.7
**Goat’s milk**	2	26.2 (9.9)	2	28.5 (6.3)
**Infant milk**	7	25.2 (14.1.)	6	17.6 (12.8)

IQR—Interquartil Range.

**Table 2 nutrients-17-03606-t002:** Adjusted iodine contents from Portuguese cow’s milk.

	Adjusted Iodine Content Median (P25, P75) (μg/100 mL)	β (95% CI)	t
**Thermic processing**			
Ultra-pasteurized *	16.4 (13.2, 17.9)	-	-
Pasteurized	20.3 (15.9, 22.2)	3.9 (−0.4, 8.1)	1.8
**Lactose presence**			
Yes *	16.4 (13.2, 17.9)	-	-
No	18.4 (17.4, 19.2)	−2.0 (−4.4, 0.4)	−1.7
**Fat level**			
Whole	17.0 (15.5, 17.7)	0.6 (−2.4, 3.6)	0.4
Semi-skimmed *	16.4 (13.2, 17.9)	-	-
Skimmed	16.9 (13.9, 20.6)	0.4 (−1.5, 2.4)	0.5
**Season**			
Winter *	16.4 (13.2, 17.9)	-	-
Summer	16.3 (14.1, 18.5)	−0.1 (−1.8, 1.7)	−0.1
**Organic and/or from pasture**			
Yes *	14.6 (9.6, 18.9)	-	-
No	16.4 (13.2, 17.9)	1.8 (−1.0, 4.7)	1.8
**Region**			
North *	22.1 (19.8, 22.8)	-	-
Centre	28.9 (18.9, 39.9)	6.8 (3.8, 9.9)	4.4
Lisbon and Tejo Valley	19.4 (15.0, 20.8)	−2.6 (−5.0, 0.6)	−1.6
Alentejo	27.5 (11.4, 20.9)	−4.6 (−7.5, −1.6)	−3.0
Azores	16.4 (13.2, 17.9)	−5.0 (−8.7, −2.7)	−3.7

* Group of comparison for the quantile regression model.

**Table 3 nutrients-17-03606-t003:** Characterization of iodine content of cow’s milk representing winter, by region.

	All Products		Organic and/or from Pasture		Not Organic or From Pasture	
	*n*	Iodine Content Median (IQR) (μg/100 mL)	*n*	Iodine Content Median (IQR) (μg/100 mL)	*n*	Iodine Content Median (IQR) (μg/100 mL)
**All regions**	**137**	19.9 (9.9)				
**Portugal**	**104**	**20.9 (9.7)**	16	16.5 (10.2)	88	21.6 (9.5)
Norte (North)	22	22.5 (7.4)	1	40.3	21	22.3 (6.5)
Centro (Center)	22	30.8 (13.5)	0	NA	22	30.8 (13.5)
Lisboa e Vale do Tejo (Lisbon and Tejo Valley)	15	20.3 (7.8)	0	NA	15	20.3 (7.8)
Alentejo	20	16.5 (8.2)	3	21.9 (13.0)	17	15.9 (6.0)
Azores	25	17.7 (4.7)	12	15.7 (8.1)	13	19.7 (6.7)
**Spain**	**31**	**15.7 (7.8)**				
**France**	**1**	**25.7**				
**Germany**	**1**	**11.4**				

**Table 4 nutrients-17-03606-t004:** Estimated contribution of milk to iodine intake in Portugal.

	Milk (g/Day) *	Iodine Milk Content (μg/100 g)	Adequate Intake (μg) **	Estimated Contribution to Iodine Intake (μg) and % of Contribution to Adequate Intake
**IAN-AF *, 2015–2016 [12]**		19.9		
Older adults (65–84 years)	156		150	31.0 (21%)
Adults (18–64 years)	149		150	29.7 (20%)
Adolescents (10–17 years)	250		120–130	49.75 (40%)
Children (<10 years)	267		90	53.1 (58%)

* IAN-AF: National Food, Nutrition and Physical Activity Survey, 2015–2016. ** EFSA, 2014 [26].

**Table 5 nutrients-17-03606-t005:** Characterization of iodine content and fortification of plant-based milk alternatives.

	Non-Enriched		Enriched with Iodine	
Plant-Based Milk Alternatives	Number of Products	Iodine Content μg/100 mL (Median, IQR)	*n* (%)	Iodine Content μg/100 mL (Median, IQR)
**All (128)**	**124**	**3.3 (5.3)**	**4 (5%)**	**24.9 (7.7)**
Soy (37)	37	3.4 (5.4)	0	-
Oat (27)	24	2.0 (4.2)	3 (11%)	26.2 (17.6)
Almond (22)	22	3.5 (4.0)	0	-
Rice (16)	16	3.0 (2.7)	0	-
Hazelnut (5)	5	1.8 (3.3)	0	-
Coconut (3)	3	7.9 (2.7)	0	-
Combinations (18)	17	4.1 (5.3)	1 (6%)	4.2

## Data Availability

The raw data supporting the conclusions of this article will be made available by the authors on request.

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
