# Peer review of "Mapping Iodine Sources for Human Nutrition in Portugal Considering Geography, Seasonality, and Processing: Milk and Plant-Based Milk Alternatives"

_nutrients, 2025, doi:10.3390/nu17223606_

Round 1

Reviewer 1 Report

Comments and Suggestions for Authors

A topic of significant health and political importance is the issue of iodine deficiency and the changing sources of iodine supply, which is crucial in the context of public health.

Broad market sampling: collecting samples from major retail chains increases the representativeness of the study.

Use of a classic analytical method: the Sandell–Kolthoff method, following appropriate digestion, is the reference method, although it requires validation.

Consideration of differentiating factors: region, season, type of milk, and technological process.

Incomplete description of method validation:
Data on repeatability, standard deviation, limit of detection (LOD), and limit of quantification (LOQ) are missing.
The authors refer to previous work on this method, but in the original publication, such data should be provided directly for each series of measurements.
It is not clear whether reference materials (certified control samples) were used.

Inaccurate description of the sampling method:
The statement “one unit of each product” is insufficient—there is no information on the number of replicates, batch numbers, or expiration dates.
It is unclear whether products of the same brand but with different production dates were treated as independent samples.
This makes it impossible to assess inter-batch variability, a key aspect in the analysis of commercial foods.

The authors use parametric tests (t-tests) despite clearly skewed distributions, as evidenced by large differences in standard deviations and medians in the tables.
There is a lack of data on repeatability, standard deviation, limit of detection (LOD), and limit of quantification (LOQ). 
The authors refer to earlier work on this method, but in the original publication, such data should be provided directly for each series of measurements. 
It is not stated whether reference materials (certified control samples) were used. 
The description of the sampling method is inadequate: 
The statement ‘one unit of each product’ is insufficient, as there is no information on the number of replicates, batch numbers, or expiry dates. 
It is unclear whether products of the same brand but with different production dates were treated as independent samples. 
This omission makes it impossible to assess inter-batch variability, which is a key aspect in the analysis of commercial foods. 
Statistics and data analysis: 
The authors use parametric tests (t-tests) despite clearly skewed distributions, as indicated by large differences between standard deviations and medians in the tables. 
There is no regression analysis or multivariate modelling to identify independent predictors of iodine content (e.g. region, type of production, processing). 
Effect sizes are reported, but without interpretation of their biological significance. 
There is a lack of standardisation in comparisons between groups: 
Comparisons between seasons or regions did not account for differences in milk types (e.g. proportion of organic products).

This may lead to incorrect conclusions regarding the absence of seasonal differences. The authors state that the lack of differences between seasons is due to ‘maintaining stable cattle feeding practices’. This claim is unsupported, as no data on feed were collected.

An alternative explanation could be the lack of sensitivity of the method or an insufficient number of comparable samples between seasons.

Overly simplistic conclusions about the role of milk:
Estimates of milk’s contribution to iodine intake are based on data from nutrition surveys conducted ten years ago (IAN-AF 2015–2016), which raises questions about their current relevance. No analysis of the uncertainty of these estimates has been performed.

Lack of comparative analysis for plant-based drinks:
The authors do not statistically distinguish between fortified and non-fortified products in their analysis; they provide only global averages. This is a significant error, as this distinction is of greatest practical importance.

Overly superficial discussion of regional factors:
The observation of lower iodine content in milk from the Azores and southern Portugal is noteworthy, but a geochemical interpretation is missing. The authors do not refer to data on soil composition, water, or feed characteristics, so the conclusion remains speculative.

The abstract is verbose and does not clearly state the study's purpose. I recommend reducing its length by one third and explicitly stating the purpose, method, main results, and practical conclusion.

Introduction: The section on the WHO and supplementation in women can be shortened, as it digresses from the main aim of the study.

Methods: The calculation method for effect size (η², d) and the interpretation thresholds should be clearly specified.

Discussion: It is advisable to relate the results more directly to practical applications. For example, could the introduction of mandatory iodine fortification of plant-based beverages realistically improve iodine status?

The technical language should be improved in several places. For example, ‘milk presented considerable iodine’ should be revised to ‘cow’s milk contained moderate-to-high iodine levels’.

The authors declare some limitations, but not sufficiently. In this version, the following should be done:

  1. Clearly indicate that the lack of data on feed, soil and water limits the interpretation of the sources of regional differences.
  2. Point out that the data only refer to retail products — they do not include raw milk, fermented milk or processed products.
  3. Emphasise that iodine content does not equal its bioavailability.
  4. Make it clear that the data on milk consumption is almost a decade old, which significantly underestimates the relevance of the estimates.

The study addresses an important topic, but its analytical credibility and interpretative power are limited by its simplified methodology and insufficient reporting of results.

The article has potential, but requires significant refinement in order for the results to make a real contribution to the scientific literature and serve as a basis for health policy.

The article may be considered for publication after significant methodological, editorial and interpretative revisions.

Key recommendations for authors:

  1. Add complete method validation data (LOD, LOQ, RSD, control materials).
  2. Reorganise the methods section: clearly specify the number of samples, how they were selected and their repeatability.
  3. Separate the analysis of fortified and non-fortified beverages.
  4. Correct tables and units; introduce consistency in formatting.
  5. Introduce multivariate analysis (e.g. ANOVA or regression) instead of only non-parametric tests.
  6. Update references on milk consumption and iodine requirements.
  7. Shorten the discussion and focus on the interpretation of results rather than repetitions of the literature.

Author Response

Comments 1:

Incomplete description of method validation:
Data on repeatability, standard deviation, limit of detection (LOD), and limit of quantification (LOQ) are missing.
The authors refer to previous work on this method, but in the original publication, such data should be provided directly for each series of measurements.
It is not clear whether reference materials (certified control samples) were used.

We thank the Reviewer for the comment. In the revised version of the manuscript (Line 112), we have added information regarding the method validation. The method LOD and LOQ were calculated according to the IUPAC guidelines. Repeatability is reported for both matrices (milk and plant-based drinks) in both sampling series (summer and winter). Certified milk matrix reference materials for iodine were not available at the time of analysis. In the absence of a matrix matched certified reference material, accuracy was assessed using a (1) in house pooled milk QC (aliquots of homogenized UHT cow milk stored frozen and processed with each batch; (2) a commercial infant formula with a declared iodine content (treated as a practical reference sample), and (3) matrix spike recoveries of 100 milk samples and 48 plant-based drink samples.

Inaccurate description of the sampling method:
The statement “one unit of each product” is insufficient—there is no information on the number of replicates, batch numbers, or expiration dates.
It is unclear whether products of the same brand but with different production dates were treated as independent samples.
We now added “Products with same brand and characteristics but from different seasons were treated as independent samples.” Line 135
This makes it impossible to assess inter-batch variability, a key aspect in the analysis of commercial foods.

Comments 2:

The authors use parametric tests (t-tests) despite clearly skewed distributions, as evidenced by large differences in standard deviations and medians in the tables. Agreed, analysis is now based on a quantile regression model.
There is no regression analysis or multivariate modelling to identify independent predictors of iodine content (e.g. region, type of production, processing). 
Effect sizes are reported, but without interpretation of their biological significance. 
Added a quantile regression model.
There is a lack of standardisation in comparisons between groups: 
Comparisons between seasons or regions did not account for differences in milk types (e.g. proportion of organic products).
Added a quantile regression model that now adjusts for the indicated variables.

This may lead to incorrect conclusions regarding the absence of seasonal differences. The authors state that the lack of differences between seasons is due to ‘maintaining stable cattle feeding practices’. This claim is unsupported, as no data on feed were collected. Presented in discussion “While there is no specific legislation regarding cattle feed in Portugal, inspection of feed labels for small farmers and industrial farmers, indicates that concentrates typically contain about 1.5 mg iodine/kg.” Line 212

An alternative explanation could be the lack of sensitivity of the method or an insufficient number of comparable samples between seasons.

The method demonstrated sufficient sensitivity to analyze both milk and plant-based drinks. The limit of detection and quantification of the method were lower than the iodine level in the samples. The standard addition approach was employed to minimize the potential matrix interference.

Comments 3:

Overly simplistic conclusions about the role of milk:
Estimates of milk’s contribution to iodine intake are based on data from nutrition surveys conducted ten years ago (IAN-AF 2015–2016), which raises questions about their current relevance. No analysis of the uncertainty of these estimates has been performed.

While the article was under revision, the recent national food balance was published (16/10/2025). We have now included this data (Line 55). As for national data on intake, a new national inquiry is still taking place.

Comments 4:

Lack of comparative analysis for plant-based drinks:
The authors do not statistically distinguish between fortified and non-fortified products in their analysis; they provide only global averages. This is a significant error, as this distinction is of greatest practical importance.

Agreed and added on table 5.

Comments 5:

Overly superficial discussion of regional factors:
The observation of lower iodine content in milk from the Azores and southern Portugal is noteworthy, but a geochemical interpretation is missing. The authors do not refer to data on soil composition, water, or feed characteristics, so the conclusion remains speculative.

Agreed and added” Lower concentrations of iodine found in milk from the Azores and Southern regions in agreement with other studies.[17,31] The Azores islands geomorphology as well as climate characteristics, irregular orography and rainfall, are known to decrease iodine environmental availability.[32]”] Line 199

Comments 6:

The abstract is verbose and does not clearly state the study's purpose. I recommend reducing its length by one third and explicitly stating the purpose, method, main results, and practical conclusion.

Agreed and altered in accordance. “Iodine is a micronutrient essential for the synthesis of thyroid hormones and crucial throughout life. Milk is potentially one of the major contributors for iodine intake in many countries, including Portugal, due to its intake patterns. This study characterizes iodine content seasonality in milk and plant-based milk alternatives commercially available in Portugal. Milk products representing summer and winter seasonal pastures and plant-based alternatives were obtained from major Portuguese supermarkets. Iodine was quantified by the Sandell-Kolthoff reaction in 146 winter and 142 summer milk products, as well as in 128 plant-based alternatives. Cow’s milk contained relevant iodine levels (Md=19.9, IQR=9.9 µg/100 ml), with no influence of thermal processing, fat, lactose content, season or being organic and/or pasture milk. However, regional differences were observed. Iodine concentration in plant-based drinks was residual, except for 4 fortified products. This study provides evidence that milk is still a relevant source of iodine in Portugal, whereas most plant-based alternatives provide negligible iodine, unless fortified. Given shifts towards plant-based diets, monitoring iodine sources and adjusting health policies to fulfil nutritional requirements are pivotal to ensure iodine adequacy.”

Introduction: The section on the WHO and supplementation in women can be shortened, as it digresses from the main aim of the study.

Thank you for the comment, we’ve altered the introduction to be more concise to the topic under study.

Methods: The calculation method for effect size (η², d) and the interpretation thresholds should be clearly specified.

To adjust for variables of impact, a quantile regression was performed where the coefficient now represents the effect and is presented in results.

Discussion: It is advisable to relate the results more directly to practical applications. For example, could the introduction of mandatory iodine fortification of plant-based beverages realistically improve iodine status?

Agreed and added “A study in United Kingdom using a dietary modelling approach demonstrated that re-placing milk for non-fortified plant-based would decrease iodine intake by up to 58% in age groups where milk represents the higher contribute; and across all age groups would increase the proportion of individuals with iodine intake below recommended levels.[43] The same study found that fortification of 22.5µg/100ml would be sufficient to prevent such decrease.[43]Line 245     

The technical language should be improved in several places. For example, ‘milk presented considerable iodine’ should be revised to ‘cow’s milk contained moderate-to-high iodine levels’.

Thank you for the comment, as there is no specific definition for what are moderate-to-high levels in milk we considered the substitution for “relevant”.

The authors declare some limitations, but not sufficiently. In this version, the following should be done:

  1. Clearly indicate that the lack of data on feed, soil and water limits the interpretation of the sources of regional differences. Added data to discussion Line 215.
  2. Point out that the data only refer to retail products — they do not include raw milk, fermented milk or processed products. Added
  3. Emphasise that iodine content does not equal its bioavailability. Added
  4. Make it clear that the data on milk consumption is almost a decade old, which significantly underestimates the relevance of the estimates. Added

 “The present work present, as a strength, the broad coverage of major national re-tailers’ assessment across seasons, and linkage of analytical results to product characteristics. As limitations, the cross-sectional nature, retail-snapshot design with potential product turnover between collection, lack of batch replication, limited representativeness of artisanal milk products (does not include raw milk, fermented milk or processed products), absence of direct linkage to individual consumption or urinary biomarkers, and reliance on the limited on-label information of some products. In addition, contributions of milk to iodine daily intake considered national intake data that is currently being revised. From the recent national food balance sheet, milk intake may be lower and, as such, the contribution of milk to iodine intake smaller than that estimated here.” Line 256

The study addresses an important topic, but its analytical credibility and interpretative power are limited by its simplified methodology and insufficient reporting of results.

The article has potential, but requires significant refinement in order for the results to make a real contribution to the scientific literature and serve as a basis for health policy.

The article may be considered for publication after significant methodological, editorial and interpretative revisions.

Thank you for your useful comments and suggestions. All were considered and altered accordingly.

Key recommendations for authors:

  1. Add complete method validation data (LOD, LOQ, RSD, control materials).

This information has now been added (Line 112).

  1. Reorganise the methods section: clearly specify the number of samples, how they were selected and their repeatability. Done.
  2. Separate the analysis of fortified and non-fortified beverages. Added.
  3. Correct tables and units; introduce consistency in formatting. Added.
  4. Introduce multivariate analysis (e.g. ANOVA or regression) instead of only non-parametric tests. Added.
  5. Update references on milk consumption and iodine requirements. Added.
  6. Shorten the discussion and focus on the interpretation of results rather than repetitions of the literature. Considered.

Reviewer 2 Report

Comments and Suggestions for Authors

The MS by Machado et al describe the iodine sources for human nutrition in Portugal: milk and
plant-based milk alternatives.

In the Materials and Methods section, the authors state that iodine concentration in milk samples was determined according to previously published methods, citing Sandell & Kolthoff (1937) and Machado et al. (2017). However, both of these references refer to the determination of iodine in urine samples, not in milk or food matrices.

The original Sandell–Kolthoff (1937) paper describes a catalytic colorimetric method based on the reduction of Ce(IV) by As(III), but it does not involve ammonium persulfate digestion. The later study by Machado et al. (2017) indeed incorporates ammonium persulfate oxidation, yet it is explicitly optimized for urine samples, addressing matrix-specific interferences relevant to biological fluids.

Therefore, the manuscript lacks a clear and adequate description or citation of a validated analytical procedure for iodine determination in milk samples.I recommend that the authors should either (a) provide a detailed description of the digestion and analytical steps actually applied to milk samples, or (b) cite a method specifically developed or validated for iodine determination in milk.

In the Abstract, the statement "Milk is potentially one of the major contributors for human iodine intake in most countries, including Portugal" is an overgeneralization.

The contribution of milk to dietary iodine intake varies markedly across regions and depends on multiple factors such as the use of iodized feed, the prevalence of iodized salt consumption, and dietary habits. While milk can be a major iodine source in some countries, this may not apply to others, where seafood and iodized salt play a more substantial role in total iodine intake.

It is suggested that the authors revise this statement to reflect regional variability.

  In Table 1 and 3,  the header "Iodine content (μg/100 mL)" is not clearly aligned,  making it ambiguous whether the unit refers to the column "N" or to "Mean, SD." This lack of alignment may confuse the reader. It is recommended that the tables be reformatted so that the unit is explicitly associated with the "Mean, SD" column, thereby improving clarity and accuracy in data presentation. 

 Τable 2  lacks a header indicating the unit of measurement for iodine content (e.g., "Iodine content (μg/100 mL)"), which is essential for data interpretation and consistency.  It is recommended to include the appropriate unit label in the column header to ensure clarity and uniformity across tables.  

 Table 5 needs to Table 5 appears to be split in the middle, and it is unclear why this is the case. Additionally, the column headers are truncated with hyphens (e.g., "vita-min"), which is not ideal. Finally, summing the numbers (N) in each row does not correspond to the values shown in the "Number of products" column.  These issues contribute to a serious problem in clarity and accuracy in data presentation.

Author Response

Comments 1: The MS by Machado et al describe the iodine sources for human nutrition in Portugal: milk and plant-based milk alternatives.

In the Materials and Methods section, the authors state that iodine concentration in milk samples was determined according to previously published methods, citing Sandell & Kolthoff (1937) and Machado et al. (2017). However, both of these references refer to the determination of iodine in urine samples, not in milk or food matrices.

The original Sandell–Kolthoff (1937) paper describes a catalytic colorimetric method based on the reduction of Ce(IV) by As(III), but it does not involve ammonium persulfate digestion. The later study by Machado et al. (2017) indeed incorporates ammonium persulfate oxidation, yet it is explicitly optimized for urine samples, addressing matrix-specific interferences relevant to biological fluids.

Therefore, the manuscript lacks a clear and adequate description or citation of a validated analytical procedure for iodine determination in milk samples. I recommend that the authors should either (a) provide a detailed description of the digestion and analytical steps actually applied to milk samples, or (b) cite a method specifically developed or validated for iodine determination in milk.

We appreciate the reviewer suggestion to justify the method applicability to milk. The Sandell-Kolthoff reaction has been widely applied to milk and dairy products and remains a practical option for food matrices (Antunes et al. 2025; Shelor & Dasgupta, 2011; Hedayati et al. 2007). In the present work we implemented a matrix-adapted Sandell-Kolthoff procedure for milk and plant-based drinks, consisting of ammonium persulfate digestion followed by catalytic colorimetry, and we validated it directly in the milk/plant-based matrices used in our study. LOD, LOQ, repeatability, between-batch precision, spike recovery percentages, are all now reported in the revised version of the manuscript (Line 107).

https://doi.org/10.1016/j.aca.2011.05.039

https://doi.org/10.1002/jcla.20185

https://doi.org/10.3168/jds.2024-25752

Comments 2:      In the Abstract, the statement "Milk is potentially one of the major contributors for human iodine intake in most countries, including Portugal" is an overgeneralization.

The contribution of milk to dietary iodine intake varies markedly across regions and depends on multiple factors such as the use of iodized feed, the prevalence of iodized salt consumption, and dietary habits. While milk can be a major iodine source in some countries, this may not apply to others, where seafood and iodized salt play a more substantial role in total iodine intake.

It is suggested that the authors revise this statement to reflect regional variability.

Agreed and added “Milk is potentially one of the major contributors for iodine intake in many countries, including Portugal, due to its intake patterns.”        

Comments 3:      In Table 1 and 3,  the header "Iodine content (μg/100 mL)" is not clearly aligned,  making it ambiguous whether the unit refers to the column "N" or to "Mean, SD." This lack of alignment may confuse the reader. It is recommended that the tables be reformatted so that the unit is explicitly associated with the "Mean, SD" column, thereby improving clarity and accuracy in data presentation.

 Τable 2  lacks a header indicating the unit of measurement for iodine content (e.g., "Iodine content (μg/100 mL)"), which is essential for data interpretation and consistency.  It is recommended to include the appropriate unit label in the column header to ensure clarity and uniformity across tables. 

 Table 5 needs to Table 5 appears to be split in the middle, and it is unclear why this is the case. Additionally, the column headers are truncated with hyphens (e.g., "vita-min"), which is not ideal. Finally, summing the numbers (N) in each row does not correspond to the values shown in the "Number of products" column.  These issues contribute to a serious problem in clarity and accuracy in data presentation.

Agreed, figures and tables altered.   

Reviewer 3 Report

Comments and Suggestions for Authors

Dear Authors

The paper “Mapping iodine sources for human nutrition in Portugal: milk and plant-based milk alternatives Interpretation depth” is extremely important for Public Health and provides a high impact directly relevant to nutrition policy and sustainable dietary transitions. Excellent balance and objectivity.

Some minor suggestions could improve the clarity and flow.

Find the detailed suggestions in the file

Author Response

Comments 1:

Lines 36–39

“Iodine is crucial for the synthesis of thyroid hormones, which regulate several metabolic pathways, throughout life, and are essential for the proper of the development of the central nervous system.”

“for the proper of the development” grammatical error.

Suggested rewrite:

“Iodine is crucial for the synthesis of thyroid hormones, which regulate numerous metabolic pathways throughout life and are essential for normal central nervous system development.”

Add a recent authoritative reference (e.g., WHO, Zimmermann MB 2023 review, DOI: 10.1017/S0029665122002762 ).

Agreed and altered.

Lines 39–41

“As such, ensuring the adequate iodine sufficiency of the population should be a public health priority.”

Could be strengthened by linking to “iodine deficiency disorders (IDDs)” to highlight clinical context.

Suggested:

“...a public health priority to prevent iodine deficiency disorders (IDDs) (Zimmermann, 2008) and associated neurodevelopmental impairment.”

Agreed and altered.

Lines 41–43

“The World Health Organization (WHO) identifies universal salt iodization as the most relevant cost-effective measure for that purpose, since it is easy to implement and adjust to salt-intake reduction policies.”

Add citation to WHO/UNICEF/ICCIDD 2023 guidance.

Change :“most relevant” “most effective and cost-efficient.”

Agreed and altered.

Lines 43–48

“In Portugal, there is no national iodine fortification program beyond the requirement for iodized salt in school canteens; which is reportedly not fully complied with. Iodized salt is therefore mostly dependent on consumers choice, representing a small proportion of all salt sales (11% in 2023). Although mandatory salt iodization was legislated for the Autonomous Region of Madeira Island in 2025, implementation is pending.”

Minor grammar: (“consumers choice” “consumers’ choice”).

Cite the primary legislative documents or surveillance reports for accuracy.

“2025” is a future date — confirm whether this refers to legislation passed in 2025 but not yet enacted or approved for enforcement starting 2025. Clarify tense (“was legislated in 2025” “has been legislated for implementation in 2025”).

Agreed and altered.

Lines 56–59

“Evidence also indicates that, in Portugal, although supplement formulations generally provide adequate iodine, supplementation duration among pregnant women is markedly insufficient, with less than 1% achieving the recommended 18-month coverage.”

Comment: clarify that “18 months” refers to pre-conception + pregnancy + lactation. Suggestion: Add short rationale for why duration is important (fetal and early infant

brain development).

Thank you for your comment. We’ve shortened the introduction and will not cover these results.

Lines 60–63

“Iodine requirements vary through the life cycle... authorities recommending 90 μg/day for preschool children...”

  • Comment: ensure references [14–17] correspond to WHO/EFSA/DRI documents.

Confirmed.

Lines 63–68

“Diet is the primary iodine intake source... Milk and dairy products often constitute a major contribution...”

  • Cite specific quantitative contributions (e.g., in Portuguese adults, milk/dairy provide X % of total iodine intake).

Agreed and added. “In the Porto region of Portugal, dairy was found to contribute to 55% to total iodine intake in working adults which is 21% of the daily iodine adequate intake [11] and a to be a relevant source in pregnant women.[12] As for milk in itself, data from the latest National Food, Nutrition and Physical Activity Survey (IAN-AF), 2015-2016, points to an intake of milk of 149g/day by adults, that would a correspond, in accordance with Scientific Update on the iodine content of Portuguese Foods, to a 24% contribution to iodine recommended intake.[13,14] Interestingly, when considering the Portuguese Food Balance Sheet, milk representativity among total food availability decreased 10.6% from the period 2016-2020 to 2020-2024.[15]” Line 48

Lines 68–73

“The European Union regulation considers a food product being a source of iodine when iodine content is equal or superior to 15% of nutrient reference value per 100 mL per serving. Mechanistically, food iodine reflects environmental iodine in soil and water, while milk iodine additionally depends on feed supplementation, on-farm hygiene practices (e.g. iodophors use), and dietary goitrogens along the dairy supply chain.”

  • Add regulation number (EU Regulation 1169/2011 or current amendment).

Confirmed and added.

Lines 79–83

“In Portugal, iodine fortification of plant-based drinks remains voluntary, with on-label iodine information inconsistently provided, further motivating surveillance of dietary iodine sources. Therefore, it is crucial to map and monitor iodine rich-foods, fortification, and supplementation strategies to adjust accordingly and prevent iodine deficiency disorders.”

  • Add quantification if available (e.g., proportion of fortified plant milks on Portuguese market).
  • Style: “iodine rich-foods” “iodine-rich foods.”
  • “To adjust accordingly” is vague — specify: “...to adjust public health policies

and food fortification programs accordingly.”

Agreed and altered. Our study was the first to identify the proportion of fortified plant-based milk, clarified. “In Portugal, iodine fortification of plant-based drinks remains voluntary, and its coverage has not been reported. In addition, the information on the product label is inconsistently provided, further motivating surveillance of dietary iodine sources.” Line 66

Lines 84–89

“Despite growing substitution of dairy with plant-based drinks, contemporary Portuguese data quantifying iodine across retail milks are limited. Therefore, in this study, we comprehensively characterized the iodine content in milk (comparing seasonality and processing characteristics), and plant-based milk alternatives available for consumers in major retailers in Portugal.”

o “quantifying iodine across retail milks” “quantifying iodine content in retail milk products.”

o Parentheses not needed; use commas: “in milk, comparing seasonal and processing characteristics, and in plant-based milk alternatives...”

Agreed and altered.

Lines 102-106

Selection Criteria : Inclusion of both cow, goat, and sheep milk in one analytic group could confound interpretation given species-related variability in iodine content. Recommend performing species-stratified analysis or clearly stating why pooled analysis was acceptable.

Agreed and altered in methods and results accordingly " To perform comparisons, only cows’ milk was included to prevent interference of species-related variability in iodine content.” Line 131

Reported data on tables changed in accordance.

Lines 107-111

Clarify whether the country of origin was determined by plant code or package labeling and whether imported UHT milk was included in “Portugal.”

Agreed and altered "Label information was recorded, comprising processing (UHT/pasteurized), fat level, lactose-free status, organic/pasture feed, nutritional panels, origin/batch codes, and any iodine related information. Country and region were attributed through the plant code of identification. Organic milk and/or from pasture were considered when reporting pasture feed or if it had biologic (organic) attribution. Line 92

Lines 112-116

Specify whether all plant-based drinks were analyzed before opening (no oxidation bias) and whether repeated measures were done for within-brand variability.

We confirm that all samples (milk and plant-based drinks) were analyzed from their original, unopened consumer packaging and within the labelled shelf life. Prior to opening, containers were inspected for integrity (no swelling, leakage, or damage seals). Immediately beforehand, containers were gently inverted to homogenize. We have clarified these procedures in the Sample collection and handling subsection.

Lines 118-126

The Sandell–Kolthoff reaction is classical but has limitations in precision compared with ICP-MS. Authors should justify its selection (e.g., cost, accessibility) and cite its method detection limit. Duplicates and analyst blinding are strong methodological points.

We fully agree that ICP-MS may offer superior instrumental sensitivity. Our choice of the Sandell-Kolthoff (S-K) method was motivated by fitness for purpose and accessibility for routine food monitoring. The iodine concentrations in our target samples are typically in ten-hundreds of µg/100 mL range. The series specific detection capability we established for our implementation of S-K (ammonium persulfate digestion, standard addition) was a LOD 0.96 µg I / 100 mL and LOQ 1.66 µg I / 100 mL, which is adequate for this concentration range.

Despite the known lower precision of colorimetry versus ICP-MS, our repeatability from duplicate pairs was 2.74% RSD (milk, summer), 3.03% RSD (milk, winter), and 3.96% RSD (plant-based, summer), which meets typical acceptance criteria for routine compositional analyses and supports the suitability of S-K for this application.

We have clarified these points in the revised version of the manuscript (Lines 105 – 125).and now report LOD and LOQ alongside repeatability, as requested.

Lines 121-124

The use of standard addition is appropriate to correct for matrix interference; however, include recovery rates and QA/QC data (e.g., certified reference materials) to ensure comparability.

We thank the Reviewer for the comment. In the revised version of the manuscript (Lines 113 – 123), we have added information regarding the method validation, including LOD, LOQ, repeatability, and certified reference materials.

Lines 154-158

No significant difference is expected—confirm statistical power adequacy (n per group) since small group sizes (e.g., n=18 for whole milk) could mask real differences.

Statistical analysis was altered and now includes a quantile regression, adjusted for the relevant variables.

Lines 159-161

Finding lower iodine in organic milk aligns with prior literature. Authors might discuss the likely mechanism (reduced mineral-supplemented feed, iodophor use). η2 = .032 indicates small effect; recommend emphasizing practical rather than statistical relevance.

Due to the new statistical method used (quantile regression) the small effect before noted is now demonstrated not having statistical significance, as now results are adjusted for other variables. Altered discussion accordingly. “Of notice, even though soils are poorer in iodine than commercial animal feed (usually fortified with iodine) no difference was found between pasture and organic milk. The definition of “organic” implies that more than 60% of the feed must be fresh or conserved forage, still the amount of animal feed used likely suppresses the differences.[30]” Line 195

Lines 162-166

Strong regional difference (η2 = 0.26) is meaningful. Suggest including map visualization for clarity. Confirm whether regional data is balanced (sample counts per region).

It is, table was formatted to be clearer. Statistical analysis was altered and now includes a quantile regression, with the coefficient corresponding to the effect size.

Lines 167-171

The approach combining national intake data with measured concentrations is methodologically sound, but uncertainty propagation (SD of both intake and content) should be shown. Values for children vs adults should specify age categories.

Agreed and included definition for age groups. There is no specification of SD or IQR measures from national intake data, the variability is only presented for dairy products overall, when presenting milk intake by age group it is not presented.

Lines 189-196

The finding that only 5% are iodine-fortified is important for public health. However, since mean iodine in fortified drinks ≈ 20 μg/100 mL, authors should discuss variability in fortification accuracy and label reliability.

We have now differentiated the values from the fortified samples, revised Table 5.

Lines 200-204

The discussion opens effectively by restating the study’s scope and major findings. However, the phrasing could be more concise (e.g., “confirms milk as a key dietary iodine source” instead of “confirms milk as a relevant source”).

Agreed and altered.

Lines 216-218

Correctly highlights goitrogenic plants. Consider specifying typical feed inclusion rates or providing references that quantify the inhibitory effect on iodine transfer to milk.

Agreed and added “Goitrogens, substances that inhibit iodine uptake by the thyroid and mammary glands, are present in several plants used in cattle feeding (e.g. canola, kale, soybean, millet, linseed) and may overestimate iodine absorption.” Line 206

Lines 219-232

Excellent mechanistic insight into iodine transfer and feed composition. However, the use of a narrative style (“on average, a cow ingests...”) could be complemented by a short quantitative schematic or equation (e.g., intake secretion efficiency).

Agreed and added “On average, a cow ingests about 6-9 kg of commercial feed per day, and roughly 30% of ingested iodine is transferred into the milk (~9 mg iodine intake →30% secretion efficiency → 14 μg/L).” Line 218

Lines 229-234

The mention of selenium is insightful. Authors might briefly mention the synergistic role of selenium and iodine in thyroid metabolism, reinforcing why dual monitoring is relevant for feed formulation

Agreed and added “Of notice, cattle feed in Portugal is also supplemented with selenium, another essential element for thyroid hormones synthesis (deiodinases are selenium dependent). Selenium deficiency can therefore exacerbate iodine deficiency.[41]” Line 223

Lines 253-258

Consider noting existing precedents in other EU countries (e.g., UK, Finland) where voluntary fortification has been introduced successfully. This strengthens the public health argument.

Agreed and added. “A study in United Kingdom using a dietary modelling approach demonstrated that re-placing milk for non-fortified plant-based would decrease iodine intake by up to 58% in age groups where milk represents the higher contribute; and across all age groups would increase the proportion of individuals with iodine intake below recommended levels.[43] The same study found that fortification of 22.5µg/100ml would be sufficient to prevent such decrease.[43]” Line 245

Lines 259-263

Recommend adding: (1) lack of batch replication, (2) limited representativeness of artisanal milk products, and (3) reliance on retail label data without verification of declared vs measured iodine.

Agreed and added point 1 and 2 in limitations. “The present work present, as a strength, the broad coverage of major national re-tailers’ assessment across seasons, and linkage of analytical results to product characteristics. As limitations, the cross-sectional nature, retail-snapshot design with potential product turnover between collection, lack of batch replication, limited representativeness of artisanal milk products (does not include raw milk, fermented milk or processed products), absence of direct linkage to individual consumption or urinary biomarkers, and reliance on the limited on-label information of some products. In addition, contributions of milk to iodine daily intake considered national intake data that is currently being revised. From the recent national food balance sheet, milk intake may be lower and, as such, the contribution of milk to iodine intake smaller than that estimated here.” Line 256

Point 3: Due to the fact that iodine content is not mandatory to be present in the nutritional table, few products presented it. Additional comparison of iodine content between measured and labelled was added to results. “In the 34 milk products reporting iodine, the median iodine content statement on food label was 20.0 μg (IQR = 0.8), and the median measured iodine content was of 23.6 μg (IQR = 14.4).” Line 162

Lines 265-268

Concise and appropriate summary. Could be strengthened by explicitly mentioning how findings could inform national iodine monitoring or future food labeling policy. Currently, it stops at “ensure iodine sufficiency” without concrete next steps.

Agreed and added “…sufficiency, not only through salt iodization but also through fortification of plant-based milk alternatives and literacy campaigns.” Line 270

Round 2

Reviewer 1 Report

Comments and Suggestions for Authors

The article addresses an important public health issue: the assessment of iodine sources in the Portuguese diet in the context of dietary changes, specifically the increased consumption of plant-based drinks at the expense of milk. The topic is of social, health, and nutritional importance, particularly given the growing trend towards plant-based diets and the associated risk of iodine deficiency.

It would also be worthwhile to relate the results more broadly to European iodine deficiency prevention programmes and to emphasise their practical significance for health policy.
The authors have justified the importance of the topic well but have not fully developed the potential of the work into practical recommendations for public health.
The objective of the study is clearly stated: to determine the iodine content in milk and plant-based drinks in Portugal, taking into account seasonality and technological processing.
However, there is no clear research question or working hypothesis, which makes it difficult to assess whether the methodology fully corresponds to the objective.
In future studies, it would be useful to clarify the objectives in the form of measurable research questions and hypotheses (e.g. ‘Are plant-based drinks a significant source of iodine compared to cow's milk?’).
The title is correct, unambiguous, and appropriate to the scope of the work. It reflects the main topic of the study (comparison of milk and its plant-based alternatives in terms of iodine content).
However, it could be clarified further regarding seasonal variation or its significance for public health, which would increase its precision and citation potential.
The article follows the correct IMRAD structure (Introduction, Methods, Results, Discussion) and is logical and clear.
The results are presented reliably, but the discussion section needs to be expanded and should refer more strongly to European studies (e.g. comparisons with countries with similar dietary patterns).
The discussion lacks consideration of limitations regarding potential measurement errors or differences between commercial brands.
The final conclusions are too general and do not follow directly from the statistical data.

Author Response

The article addresses an important public health issue: the assessment of iodine sources in the Portuguese diet in the context of dietary changes, specifically the increased consumption of plant-based drinks at the expense of milk. The topic is of social, health, and nutritional importance, particularly given the growing trend towards plant-based diets and the associated risk of iodine deficiency.

It would also be worthwhile to relate the results more broadly to European iodine deficiency prevention programmes and to emphasise their practical significance for health policy.

The authors have justified the importance of the topic well but have not fully developed the potential of the work into practical recommendations for public health.

Thank you. We have now revised the discussion section, including not only additional comparisons with other countries but also highlighting the relevance of the findings to support public health measures.

The objective of the study is clearly stated: to determine the iodine content in milk and plant-based drinks in Portugal, taking into account seasonality and technological processing.
However, there is no clear research question or working hypothesis, which makes it difficult to assess whether the methodology fully corresponds to the objective.
In future studies, it would be useful to clarify the objectives in the form of measurable research questions and hypotheses (e.g. ‘Are plant-based drinks a significant source of iodine compared to cow's milk?’).

Thank you, added at the end of the introduction “The present study comprehensively characterized the iodine content in milk, comparing seasonality and processing characteristics, and plant-based milk alternatives available for consumers in major retailers in Portugal. Furthermore, it intended to estimate the contribution of milk for iodine intake.”

The title is correct, unambiguous, and appropriate to the scope of the work. It reflects the main topic of the study (comparison of milk and its plant-based alternatives in terms of iodine content).
However, it could be clarified further regarding seasonal variation or its significance for public health, which would increase its precision and citation potential.

Altered to “Mapping iodine sources for human nutrition in Portugal considering geography, seasonality and processing: milk and plant-based milk alternatives”

The article follows the correct IMRAD structure (Introduction, Methods, Results, Discussion) and is logical and clear.
The results are presented reliably, but the discussion section needs to be expanded and should refer more strongly to European studies (e.g. comparisons with countries with similar dietary patterns).

Thank you, added to the discussion “Similar results on the contribution of milk to iodine intake were found in Norway, United Kingdom and Ireland.[42] In addition, the proportion of iodine fortification in plant-based alternatives in these countries is also low.[43,44]”

The discussion lacks consideration of limitations regarding potential measurement errors or differences between commercial brands.

Agreed, added to the discussion the limitations regarding potential measurement errors “The present work present, as a strength, the broad coverage of major national re-tailers’ assessment across seasons, and linkage of analytical results to product character-istics. As limitations, the cross-sectional nature, retail-snapshot design with potential product turnover between collection, lack of batch replication and limitations related to the measurement method itself, limited representativeness of artisanal milk products (does not include raw milk, fermented milk or processed products), absence of direct linkage to individual consumption or urinary biomarkers, and reliance on the limited on-label information of some products.”

Regarding differences between commercial brands, as we collected all marketed products and characteristics, differences are considered in analysis.

The final conclusions  QM are too general and do not follow directly from the statistical data.

Agreed, added to the conclusion “In summary, the present study maps iodine content and variability of milk and plant-based beverage sources in Portugal. These findings provide scientific knowledge to support public health authorities to decide on the most adequate measures to ensure iodine sufficiency in the population, not only through the implementation of salt iodization but also through fortification of plant-based milk alternatives, since its iodine content is low.”

Reviewer 2 Report

Comments and Suggestions for Authors The authors do not clearly and in detail report the laboratory method they used.

Author Response

The authors do not clearly and in detail report the laboratory method they used.

Text revised “The Sandell-Kolthoff kinetic colorimetric method has been widely used for iodine determination in milk and dairy products[22-24], and was selected for its accessibility and throughput in routine monitoring laboratories. Iodine content was assessed by a matrix-adapted procedure [25,26], following Machado et al. [25,26]. Samples were digested with 1M ammonium persulfate digestion at 100 °C in a water bath for 1 hour. After reaching room temperature, arsenious acid reagent (1 M H2SO4, 0.43 M NaCl, 0.05 M As2O3) was added at a 2.5:1 (v/v) reagent:digest:sample ratio and allowed to react for 15 min prior to the addition of cerium (IV) solution (76 mM; 1:11.5 (v/v) ratio relative to mixture solution). Colour development was read exactly 30 min after the addition of cerium (IV) at 420 nm. A standard addition approach was employed to correct for potential matrix effects. For each analytical run, a six-point external calibration curve in water (0 – 468 µg I /L; 0 – 46.8 µg I / 100mL) was prepared. In parallel, a matrix-matched standard-addition curve using the same nominal iodine standards was constructed for every sample. The iodine concentration of the sample was obtained from the x-intercept of the sample standard addition regression. The method limit of detection (LOD) and limit of quantification (LOQ) were 0.96 µg I / 100 mL and 1.66 µg I / 100 mL, respectively. Within-run repeatability, estimated from duplicate pairs, was 2.74% RSD for milk in the summer series and 3.03% RSD for milk in the winter series. For plant-based drinks the RSD was 3.96%. Between batch precision, estimated from an in-house quality control material (pooled UHT cow’s milk aliquots) was analysed in every batch; the RSD across batches was 2.18%. The method accuracy was assessed using a commercial infant formula with a declared iodine content of 15 µg I /100 mL (treated as a practical reference sample). The recovery of this milk was 98 ± 2% (n = 8). In addition, 100 milk samples and 48 plant-based drink samples were spiked at low and mid-levels within the calibration range and yielding recoveries of 94 – 109%. The analyst was blinded to the product characteristics.”

Round 3

Reviewer 2 Report

Comments and Suggestions for Authors

While the digestion conditions described by the authors are consistent with published Sandell–Kolthoff protocols for urine, the approach used for milk typically requires different—often stronger—digestion or ashing procedures. In the Machado et al. studies cited by the authors, ammonium persulfate digestion was validated specifically for urine matrices; these papers do not describe or validate the method for milk or dairy products. Because milk contains higher levels of fat, protein, and organically bound iodine, it presents greater analytical challenges compared to urine. For this reason, published Sandell–Kolthoff methods for milk commonly rely on acidic or more intensive digestion protocols, such as the preparation of an acidic digestion reagent (e.g., ammonium metavanadate dissolved in perchloric acid), or other matrix-appropriate pre-treatments.

Two relevant examples demonstrating the distinct digestion requirements for milk include:

1. Hedayati M, Ordookhani A, Daneshpour MS, Azizi F. Rapid acid digestion and simple microplate method for milk iodine determination. J Clin Lab Anal. 2007;21(5):286-92. doi: 10.1002/jcla.20185. PMID: 17847102; PMCID: PMC6649143. 2. Vance KA, Makhmudov A, Shakirova G, Roenfanz H, Jones RL, Caldwell KL. Determination of iodine content in dairy products by inductively coupled plasma mass spectrometry. At Spectrosc. 2018 May 1;39(3):95-102.

Author Response

Answer to Reviewer 2

We appreciate the reviewer thoughtful comment regarding digestion strength for milk matrices. We agree that milk presents different analytical challenges than urine because of fat, protein and organically bound iodine, and that several Sandell-Kolthoff applications for milk have used harsher treatments (e.g. acidic vanadate/perchloric acid or ashing). In this study, we adopted 1 M ammonium persulfate digestion at 100 °C for 60 min followed by colour development, and validated the procedure in the milk and plant-based matrices analyzed. Multiple lines of evidence support that this pretreatment is fit-for-purpose for our samples:

  1. Matrix-corrected quantification: Every sample was quantified by per sample standard addition, which corrects any residual matrix-dependent bias in sensitivity.
  2. Accuracy in-matrix: Trueness was verified with a commercial infant formula (label 15 µg I/100 mL; 98 ± 2%, n=8) and by spike recovery across 100 milk and 48 plant-based samples (94 – 109%). Repeatability from duplicates was 2.74 – 3.96% RSD and the in-house pooled UHT milk quality control showed 2.18% RSD between batches.
  3. Detection capability adequate for purpose. We report LOD/LOQ which are appropriate for tens-hundreds of µg/100 mL range observed in these products.
  4. In addition, we performed a side-by side comparison between ammonium persulfate and acidic vanadate [Hedayati et al. 2007] digestion with subsequent Sandell-Kolthoff protocol on a subset of milk/plant-based samples. Results from the two pre-treatments were concordant, with a between-protocol RSD < 10% and no indication of systematic bias. Given the comparable performance and the safer reagent profile, we retained ammonium persulfate digestion.

To address the reviewer request for clarity, we have explicitly stated in Methods that our procedure is a matrix-adapted Sandell-Kolthoff implementation, provided reagent compositions and v/v addition ratios, clarified the standard addition per samples and added our quality control criteria. We have also added citations to milk specific Sandell-Kolthoff applications and a brief sentence acknowledging that alternative, more aggressive pre-treatments exist, while explaining why ammonium persulfate was selected here (fitness for purpose, hazardous-reagent avoidance, and demonstrated in-matrix performance).

Iodine analysis

Iodine content was assessed by a matrix-adapted Sandell-Kolthoff procedure [22],  essentially following Machado et al. [23]. Briefly, samples were digested with 1M ammonium persulfate digestion at 100 °C in a water bath for 1 hour. After reaching room temperature, arsenious acid reagent (1 M H2SO4, 0.43 M NaCl, 0.05 M As2O3) was added at a 2.5:1 (v/v) reagent:digest sample ratio and allowed to react for 15 min prior to the addition of cerium (IV) solution (76 mM; 1:11.5 (v/v) ratio relative to mixture solution). Colour development was read exactly 30 min after the addition of cerium (IV) at 420 nm. The Sandell-Kolthoff kinetic colorimetric method has been widely used for iodine determination in milk and dairy products[24][25][26], and was selected for its accessibility and throughput in routine monitoring laboratories. A standard addition approach was employed to correct for potential matrix effects. For each analytical run, a six-point external calibration curve in water (0 – 468 µg I /L; 0 – 46.8 µg I / 100mL) was prepared. In parallel, a matrix-matched standard-addition curve using the same nominal iodine standards was constructed for every sample. The iodine concentration of the sample was obtained from the x-intercept of the sample standard addition regression. The method limit of detection (LOD) and limit of quantification (LOQ) were 0.96 µg I / 100 mL and 1.66 µg I / 100 mL, respectively. Within-run repeatability, estimated from duplicate pairs, was 2.74% RSD for milk in the summer series and 3.03% RSD for milk in the winter series. For plant-based drinks the RSD was 3.96%. Between batch precision, estimated from an in-house quality control material (pooled UHT cow’s milk aliquots) was analysed in every batch; the RSD across batches was 2.18%. The method accuracy was assessed using a commercial infant formula with a declared iodine content of 15 µg I /100 mL (treated as a practical reference sample). The recovery of this milk was 98 ± 2% (n = 8). In addition, 100 milk samples and 48 plant-based drink samples were spiked at low and mid-levels within the calibration range and yielding recoveries of 94 – 109%. Because milk and plant-based drinks contain lipids and proteins that can interfere with catalytic colourimetry, alternative pre-treatments reported for milk (e.g. acidic vanadate/perchloric acid; Hedayati et al. 2007) were considered. Ammonium persulfate digestion was selected for fitness-for-purpose and reagent safety, and its adequacy is supported by the in-matrix validation data presented. However, very high-fat or strongly fortified specialty milks may require more aggressive pre-treatments; in such cases, verification of recovery and precision is recommended. The analyst was blinded to the product characteristics.”